# A357 Alloy by LPBF for Industry Applications

**DOI:** 10.3390/ma13071488

**Published:** 2020-03-25

**Authors:** Massimo Lorusso, Francesco Trevisan, Flaviana Calignano, Mariangela Lombardi, Diego Manfredi

**Affiliations:** 1Center for Sustainable Futures Technologies CSFT @Polito, Istituto Italiano di Tecnologia, Via Livorno 60, 10145 Torino, Italy; diego.manfredi@polito.it; 2Wärtsilä Italia S.p.A, Bagnoli della Rosandra 334, 34018 San Dorligo della Valle (TS), Italy; francesco.trevisan1990@gmail.com; 3DIGEP–Dipartimento di Ingegneria Gestionale e della Produzione, Corso Duca degli Abruzzi 24, 10129 Torino, Italy; flaviana.calignano@polito.it; 4DISAT–Dipartimento di Scienza Applicata e Tecnologia, Politecnico di Torino, Corso Duca degli Abruzzi 24, 10129 Torino, Italy; mariangela.lombardi@polito.it

**Keywords:** laser powder bed fusion (LPBF), A357, heat treatment, solution, ageing

## Abstract

The aim of this study is to define the process parameters to build components for industrial applications in A357 alloy by Laser Powder Bed Fusion (LPBF) and to evaluate the effects of post-processing heat treatments on the microstructure and mechanical properties in order to obtain the highest hardness and strength. First, process parameters values were defined to obtain full dense components with highest productivity. Then samples were built for microstructural, hardness, and tensile strength investigation in different conditions: as-built, after a stress-relieving treatment, and after a T6 precipitation hardening treatment. For this latest treatment, different time and temperatures for solution and ageing were investigated to find the best in terms of final hardness achievable. It is demonstrated that samples in A357 alloy can be successfully fabricated by LPBF with a density of 99.9% and a mean hardness value achievable of 116 HV0.1, in as-built condition. However, for production purposes, it is fundamental to reduce the residual stresses typical of LPBF. It was shown that a similar hardness value could be obtained after a stress-relieving treatment followed by a proper T6 treatment, together with a coarser but more isotropic microstructure.

## 1. Introduction

The growing demand for materials suitable for critical structural applications in the automotive and aerospace fields have promoted the development of Al–Si–Mg alloys, a group of heat-treatable cast Al–Si alloys, on account of their several advantages including excellent casting characteristics, good corrosion resistance and weldability, and high strength-to-weight ratio in heat-treated condition [1,2,3,4,5]. In recent years, these alloys have become particularly attractive for the production by rapid solidification techniques, such as melting spinning [6], which display the advantage of refined microstructure and extended solid solubility. Laser Powder Bed Fusion (LPBF), a powder bed-based additive manufacturing (AM) process for metal, has attracted great interest for the possibility to produce full dense near-net-shape metallic structures with complex geometries combined with extremely high cooling rates, which generate ultra-fine microstructures [7,8,9,10,11] or even metastable phases [12], such as bulk metallic glasses [13]. Because of the short interaction times and highly localized heat input, large thermal gradients exist during the LPBF process [14].

Thermal stresses generate residual stresses inside the material which can cause distortion or warping of the component produced [15,16,17]. To reduce this problem, components simple in shape, such as cubes or tensile test samples, were generally fabricated attached to the building platform made by a similar alloy, such as extruded parts and therefore they are then cut by Electrical Discharge Machining (EDM). Parts with complex geometry are usually fabricated with support structures necessary to fix them to the building platform, and in particular to sustain protruding surfaces which otherwise could collapse, to conduct excess heat away from them and to prevent their warping. After the LPBF process for samples, with or without supports, it is fundamental to perform stress a heat treatment to relieve the residual stresses. The geometrical design and optimization of these structures were studied to improve the sustainability and efficiency of metallic parts produced by LPBF [18].

Moreover, in the LPBF process, it is fundamental to optimize the main process parameters, such as laser power and scanning speed, layer thickness, hatching distance, scanning strategy, with respect to the starting powder in order to obtain dense components. In the literature, the influence of the different process parameters on the microstructure and mechanical properties [17,19,20,21,22,23], density and surface quality [24,25,26,27,28,29] has been investigated. Other research has focused on simulation of the melt pool behavior [30], on kinetics [31], on post-processing treatments [32,33,34], on corrosion behavior [35,36] and on tribology properties [37,38]. However, the processing parameters need to be correlated with the resulting mechanical properties. Buchbinder et al. [39] have investigated and correlated the process parameters on the microstructure and resultant mechanical properties for AlSi10Mg alloy. Siddique et al. [40] and Sercombe et al. [41] investigated the effect of the energy density, build plate preheating, and post-processing stress relief on the tensile properties of LPBF AlSi12, reporting that the energy density had the strongest influence.

Among Al–Si–Mg alloys, A357 or AlSi7Mg0.6 is a cast alloy extensively used in structural automotive applications such as blocks, cylinder heads, suspension system etc. since its very good castability and the specific strength, good corrosion and fatigue resistance [42]. Generally, A357 is strengthened by precipitation hardening [43] particularly T6 heat treatment [44] which consists of solutionizing at high temperature, subsequent water quenching and final artificial ageing. The optimum castability of such alloy is provided by Si presence which increase the fluidity and hot cracking resistance, while the Mg presence improves the mechanical properties after T6: during heat treatment Mg and Si form Mg_2_Si which precipitates from α-Al solid solution and strengthens the alloy precipitation hardening [45,46]. Few studies have been addressed on the production and characterization of A357 parts made by LPBF. Rao et al. [47] and Aversa et Al. [14] investigated the effects of laser parameters and building platform temperature on density and mechanical properties of LPBF A357 parts. Casati et al. [48] and Yang et al. [49] investigated the A357 alloy produced by LPBF to evaluate its response to thermal ageing. Oliveira de Menezes et al. [50] studied the effect of build orientation on tensile behavior of A357 Al alloy produced by LPBF.

The focus of the present study was strictly connected to industrial needs. Therefore, it was decided to study systematically the effect of different process parameters for A357 alloy, with the aim of reaching the highest density with the highest productivity. Furthermore, the stress-relieving treatment was considered a fixed point. In fact, in the production of a complex part, the real advantage given by AM technologies, fundamental is to release the thermal stresses before detaching it from the building platform, to avoid distortion or loss of dimensional tolerances. In many cases, it would be difficult or impossible to treat the component still attached to the building platform. Moreover, detaching the part from the building platform allows efficient evacuation of the residual powder entrapped into cavities during the process. Therefore, in this study also the systematic investigation of a T6 treatment to reach the highest hardness, was performed after the stress-relieving treatment.

## 2. Materials and Methods

Specimens were fabricated using an EOSINT M270 Dual mode machine, equipped with 200W Yb fiber continuous laser beam with a focused diameter of 100 µm. The specimens were produced from a gas atomized powder of A357 (AlSi7Mg0.6) aluminum alloy provided by LPW Technology Ltd. (Runcorn, United Kingdom). A preliminary observation of powder morphology was conducted using a Field Emission Scanning Electron Microscope (FESEM) Zeiss SupraTM 40 (Oberkochen, Germany), as shown in Figure 1a. The graph of Figure 1b illustrates the particle size distribution with volumetric assumption, obtained by laser diffraction (Fritsch model Analysette 22 Compact). Using this assumption, the mean diameters corresponding to 10% (d10), 50% (d50) and 90% (d90) of the cumulative size distribution are 22, 34 and 53 µm, respectively.

Cubic samples of 10 mm per side were fabricated in Ar atmosphere, with residual oxygen concentration below 0.10%, by varying the scanning speed and hatching distance [25] within the ranges shown in Table 1 at constant layer thickness of 30 µm and with laser power of 195 W. The building platform temperature was kept at 100 °C. The scanning strategy adopted consisted of 5 mm stripes rotated of 67° among consecutive layers [51]. This should ensure a better overlapping and more isotropic properties with respect to other scanning strategies made of layers with unidirectional stripes or at least with a cross-ply pattern.

Density values were evaluated based on Archimedes method [28] according to standard ASTM B962-14, using an ORMA BCA 200S balance (accuracy ±0.1 mg). On the other hand, the productivity of the LPBF process is given by the process-related build-up rate, in according to Equation (1) [52].
Prod_LPBF_ = h_d_ × t × v(1)
where h_d_ is the hatching distance [mm], t is the layer thickness and v is the scanning speed [mm/s].

To define the optimized process parameters for A357 alloy, a combination of the highest relative density and productivity was considered.

Once defined such parameters, the effects of two different post-process heat treatments on the microstructure, hardness, and tensile properties of A357 specimens by LPBF were investigated. Therefore, other cubic samples and cylindrical bars parallel to the building plane, with a diameter of 4 mm and gage length 20 mm for tensile testing according to ASTM E8/E8M-09, were fabricated. The samples characterized directly after the process were defined as-built to distinguish them from those characterized after heat treatments. The first one, the stress-relieving treatment, was performed at 300 °C for 2 h in air to release the high residual stresses induced during LPBF manufacturing [16,53,54]. Such treatment usually enhances the ductility and reduces the tensile strength of the LPBF samples with respect to the as-built conditions [51]. Therefore, in order to try to enhance again strength and hardness, the second heat treatment was a T6, including solution, water quenching, and artificial ageing, well known in the literature with Al–Si–Mg alloys for precipitation strengthening [6,43,45]. Different T6 treatments were investigated and carried out, as summarized in Figure 2: two distinct annealing solutions, Solution A performed at 530 °C for 5 h and Solution B performed at 540 °C for 8 h, each of them followed by ageing at 160 °C or at 170 °C, with different duration times, from 1.5 to 12 h.

To evaluate the hardness of samples, ten micro-Vickers indentations were made on as-built and each heat-treated cubic samples, with a load of 100 g for 15 s, according to standard EN ISO 6507-1, using Leica VMHT microindenter. Then microstructural investigations were carried out by means of optical and electron microscopy (FESEM) on as-built, stress-relieved, and the best T6 specimens in terms of hardness. The chemical analysis was also performed by X-ray diffraction (XRD) using X-Pert Philips diffractometer (PANalytical, Almelo, The Netherland). Scherrer method was used to evaluate crystallite dimensions.

Microstructural characterization specimens were first polished with 1 µm diamond paste and by sub-micrometric colloidal silica suspension, then etched for 12 s with Keller’s reagent. The surfaces analyzed were taken from sections parallel to the building direction.

Finally, tensile tests were performed on samples in the same three heat treatment conditions, through a Zwick Z100 machine at room temperature. These tests were performed adopting 35 N pre-loading and a 0.008 s^−1^ deformation rate. Fracture surfaces were observed with FESEM.

## 3. Results and Discussion

### 3.1. Optimization of Process Parameters

In Table 2, LPBF process parameters, productivity, and density values of A357 samples are summarized. Relative density of specimens is calculated assuming a theoretical value of 2.68 g/cm^3^. Considering productivity, the main influencing variables to decrease the time required by laser beam to melt the powder layer and thus, manufacture parts economically, are hatching distance (h_d_), layer thickness (t) and scanning speed (v). The layer thickness and scanning speed are limited among other factors by the available laser power. The hatching distance was limited by the diameter of the beam.

Starting from the assumption of getting parts with density greater than 99.5% and high productivity, it is considered to adopt the parameters of sample 5 to produce the specimens for the mechanical tests.

### 3.2. Evaluation of Micro-Hardness

Figure 3 shows the results for the different T6 treatments. Focusing on the ageing time influence, it can be noticed that for the Solution B, samples over-ageing starts after 3 h, instead for the Solution A the over-ageing begins only after 5 h. Considering the temperature, ageing at 160 or 170 °C seems to not affect the peak hardening conditions, while on the contrary, a slight difference can be observed in the pre-aged and over-aged conditions.

Vickers micro-hardness mean values are summarized in Table 3. As expected, the highest hardness is obtained in the as-built conditions, so without treatment, while the lowest value was obtained after the stress-relieving. Analyzing in detail the results of different T6 treatments, it appears that a maximum hardness of 116 ± 2 HV_0.1_ is reached for samples T6_B,170,3_. In the case of no ageing conditions, the mean hardness for the 8 h solution treated was 82 ± 2 HV_0.1_, 14% higher than the ones aged for 5 h, while after 12 h ageing the maximum hardness reached is 112 ± 2HV_0.1_.

All the data provided give evidence of a difference between the two solution treatments: the great improvement given by longer solution treatments to hardness rather than the shorter ones. As described by Aboulkhair et al. [55] in the case of precipitation hardening treatment material strengthening is governed by the presence of precipitates (Orowan strengthening), solid solution strengthening, and dislocation strengthening. As reported in the literature, in the case of cast 357, fine Si particles and Mg precipitates, acting as obstacles to the dislocation motion in peak hardening conditions. On the other hand during over-ageing Orowan strengthening effect is depressed and hardness is reduced. This can be observed for both T6_A_ and T6_B_ at 12 h ageing time (Figure 3).

From the micro-hardness measurements it can be observed that prolonging the solution treatment from 5 h to 8 h considerably varied the material behavior in the early stages of ageing and in the peak hardening conditions. Longer solution treatments differently from shorter ones favor higher diffusion of Si inside the microstructure, completely homogenizing the LPBF microstructure and enhancing material micro-hardness [55]. This behavior appears to be opposite to the typical cast parts treatments, where shorter solution treatments are preferred to longer ones [55]. These statements are consistent with previous literature: as stated by Leuders et al. [56], LPBF produces very fine microstructure that require adjustments to the post-process heat treatments traditionally adopted on cast parts.

### 3.3. Microstructure Investigation

Figure 4 and Figure 5 show the microstructures of A357 alloy samples, treated in different conditions, as observed through optical and electronic microscopes. The typical LPBF microstructure made of superimposing melt pools created by the laser scan tracks can be appreciated in the as-built specimens (Figure 4a) and in the stress-relieved ones (Figure 4b). As can be appreciated at high magnification (Figure 5a,b), and in agreement with recent literature, the microstructure is very fine in both conditions. From XRD analyses (Figure 6) the dimension of Al and Si crystallites is calculated by the Scherrer Method. For as-built materials the mean crystallite size is 51 nm for Al and 16 nm for Si, while for samples after stress-relieving treatment the mean size slightly increases, being 60 nm for Al and 24 nm for Si. On the other hand, XRD spectra did not reveal any peak for Mg_2_Si intermetallic compound.

Considering the effects of newly defined T6 treatment, T6_B,170,3_, upon A357 alloy by LPBF, optical micrograph gives evidence of melt pools disappearing (Figure 4c). At higher magnification (Figure 5c and Figure 7), it could be stated that microstructure is now made of homogenous distribution of free Si in the form of micrometric precipitates surrounded by α-Al phase.

During LPBF process, the alloy melt cools down very rapidly, with mean rate of 10^3^–10^5^ K/s [21], creating an anisotropic and ultra-fine microstructure. During the rapid cooling, the α-Al solidifies first in the preferential cellular structure, rejecting the residual Si from the solid solution to the grain boundaries [32]. Typically, Al microstructures by LPBF can be divided in fine and a coarse cellular structure inside the melt pool and heat affected zone around the melt pool in the previously deposited layers [51]. Post-process stress-relieving treatment reduces the residual stresses generated inside the material, slightly modifying the microstructure of the material as shown by XRD analysis (Figure 6). On the other hand, T6 treatment totally modifies the microstructure:Si cristallites and eutectic (α + Si) lamellar phases disappeared leading to α-Al matrix and Si micro-precipitates, thanks to the higher Si diffusion rate inside the α-aluminum solid solution [57].

### 3.4. Evaluation of Tensile Properties

The mechanical properties of A357 sample are summarized in Table 4, while the stress–strain curves are shown in Figure 8. As expected, it can be observed that all post-treatments altered the tensile properties with respect to the as-built conditions. Stress-relieving treatment strongly affects yield strength (Rp,0.2) and ultimate tensile strength of LPBF parts, reducing their values of about 30 and 34%, respectively.

On the contrary, elongation at break increases of about 36%. Moreover, T6_B,170,3_ treatment improves the tensile strength of A357 samples after stress-relieving. The *R_p_*_,0.2_ mean value rises up to 249 MPa, similar to the one obtained before any post-treatment. At the same time, ductility decreased after T6, reaching values comparable to the as-built conditions.

The fracture surfaces of different samples were analyzed by FESEM and some significant micrographs are reported in Figure 9. It could be assumed that all LPBF samples exhibited trans-granular ductile fracture: fine dimples structure, which is the consequence of plastic deformation, can be noticed in all images [32]. Moreover, at higher magnifications (Figure 9b,d,f) it can be noticed how the heat treatments cause the dimples growth. Dimples dimensions vary from hundreds of nanometers in the as-built conditions, to sub-micrometric dimensions after annealing up to microns after T6_B,170,3_.

## 4. Conclusions

The aim of this study is to define LPBF main process parameters to build dense samples with the highest productivity in A357 alloy, and to perform then the proper post-process heat treatment, in order to have the highest hardness without residual thermal stresses, useful for industrial components in such alloy.

It was demonstrated that using a EOSINT M270 machine with a 200 W laser, it was possible to obtain A357 samples with a relative density higher than 99.5% and a production rate of about 4.1 mm^3^/s. Tensile tests show that a stress-relieving treatment at 300 °C for 2 h reduces micro-hardness and tensile strength and increases the ductility of with respect to the as-built parts. To improve the mechanical properties, a subsequent T6 treatment was investigated through 2 solution treatments 530 °C for 5 h, called Solution A, and 540 °C for 8 h, called Solution B., followed by 2 ageing treatments, at 160 °C or 170 °C for 1.5, 3, 5, 8, or 12 h. The ageing curves revealed a peak hardening for Solution B followed by ageing at 170 °C for 3 h, being the mean micro-hardness value obtained 116 HV, comparable to as-built mean value 119 HV. This T6 treatment had also increased the yield strength, reaching values similar to those of as-built samples, leading to a more isotropic material, without melt pools. Microstructural and fracture surfaces analyses show the dimples growth after T6 heat treatment, confirming agreement with the higher ductility of the A357 alloy.

## Figures and Tables

**Figure 1 materials-13-01488-f001:**
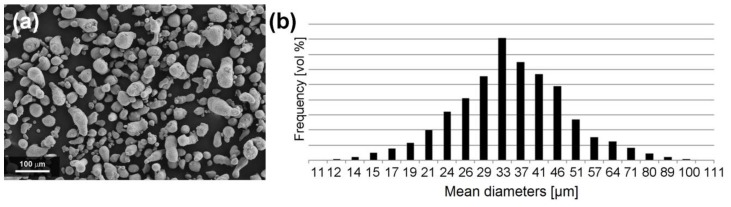
A357 alloy powder: (**a**) FESEM observation (**b**) laser granulometry analysis.

**Figure 2 materials-13-01488-f002:**
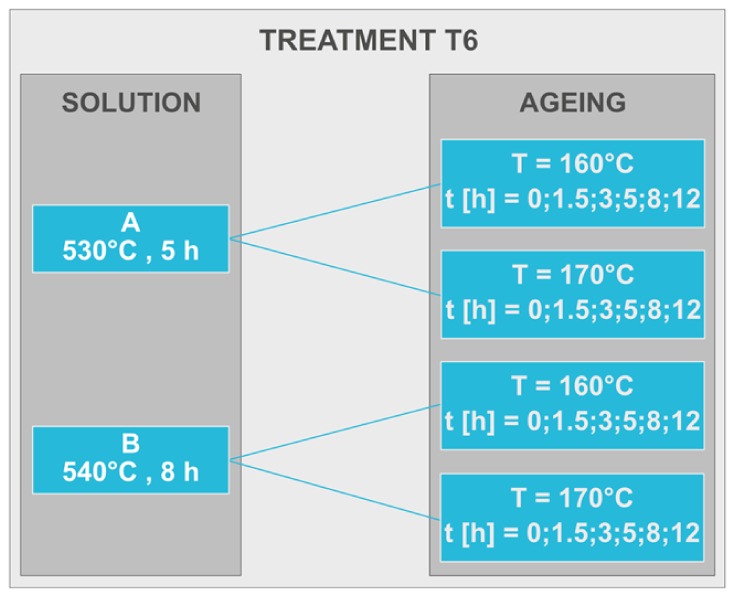
The different T6 treatment conditions investigated on A357 alloy by LPBF.

**Figure 3 materials-13-01488-f003:**
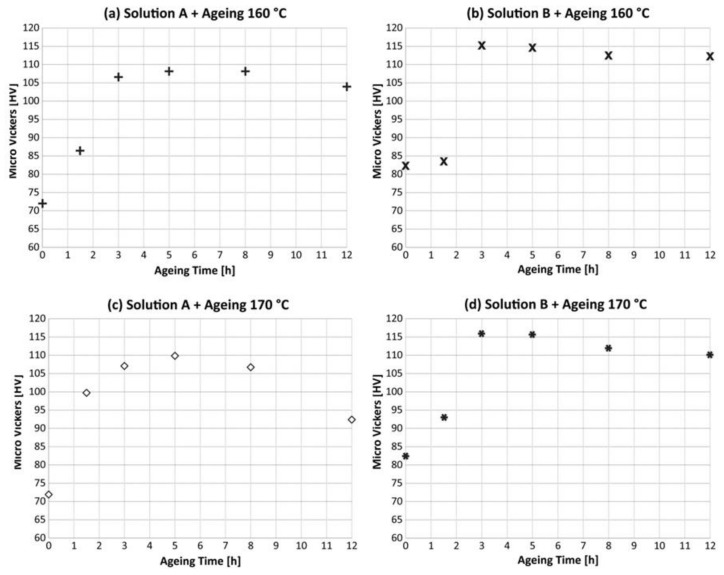
Vickers micro-hardness (HV_0.1_) mean values obtained for samples after different T6 conditions.

**Figure 4 materials-13-01488-f004:**
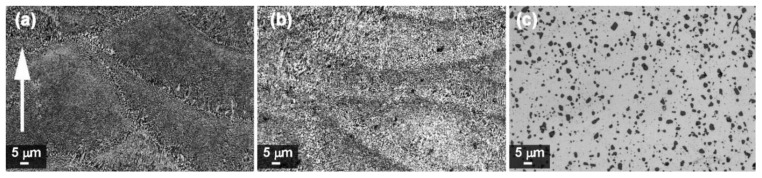
Optical images of A357 samples: (**a**) as-built; (**b**) after stress-relieving; (**c**) after T6_B,170,3_ heat treatment. The white arrow indicates building direction.

**Figure 5 materials-13-01488-f005:**
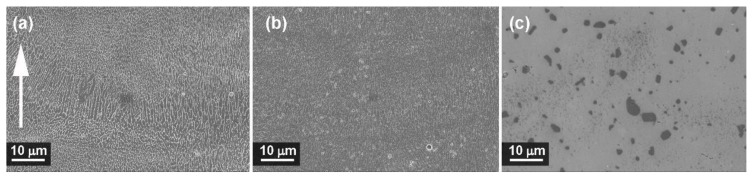
FESEM images of A357 samples: (**a**) as-built; (**b**) after stress-relieving; (**c**) after T6_B,170,3_ heat treatment. The white arrow indicates building direction.

**Figure 6 materials-13-01488-f006:**
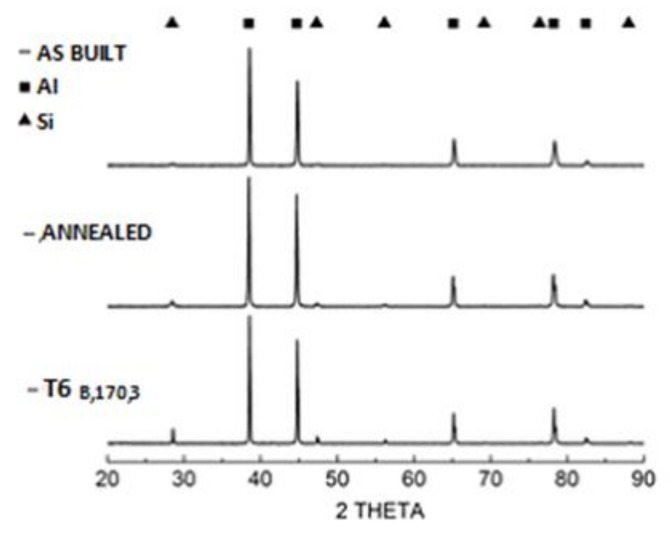
XRD patterns of A357 alloy samples.

**Figure 7 materials-13-01488-f007:**
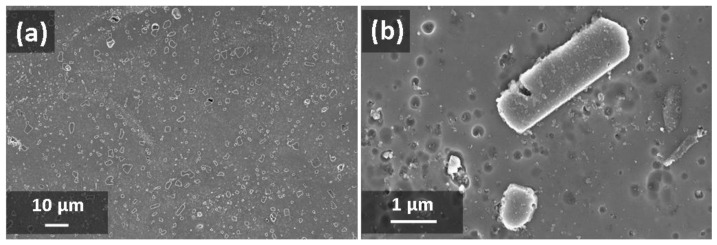
FESEM images of etched microstructure of A357 specimens produced by LPBF, after T6_B,170,3_ heat treatment: overall microstructure (**a**) and MG_2_Si precipitates (**b**).

**Figure 8 materials-13-01488-f008:**
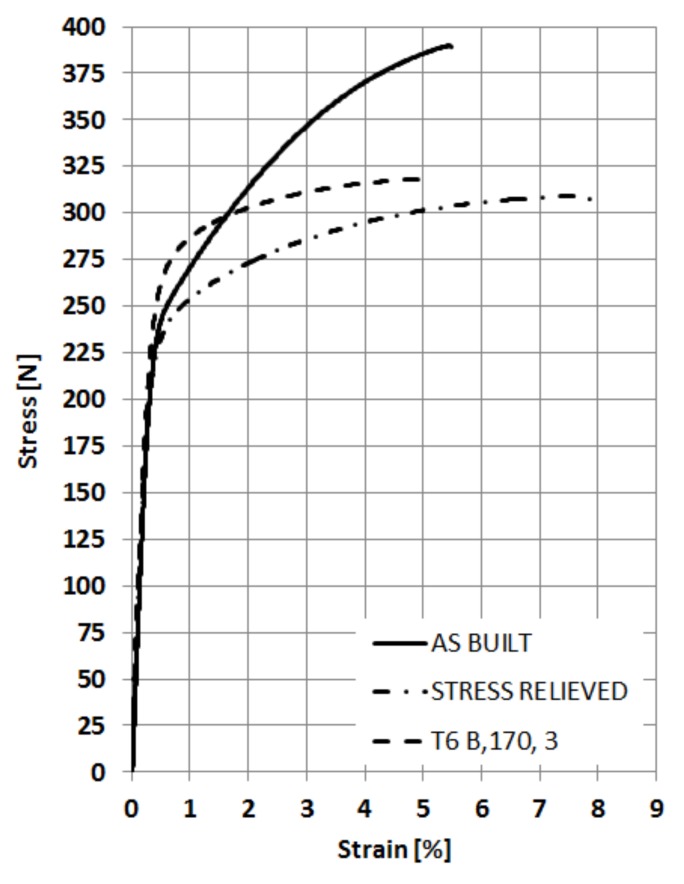
Stress–strain curves of room temperature tensile tests performed on A357 specimens by LPBF in different conditions.

**Figure 9 materials-13-01488-f009:**
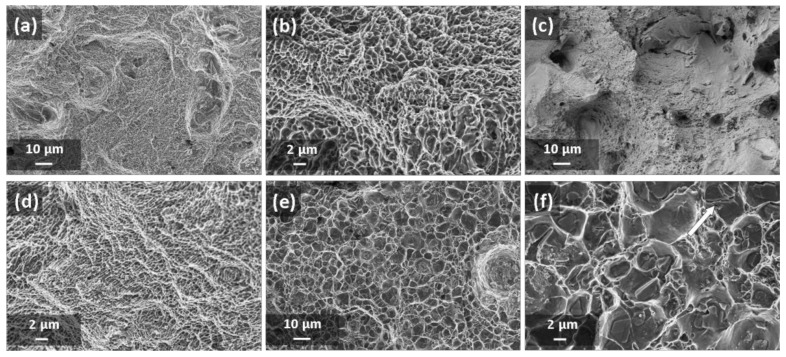
Fracture surfaces analysis through FESEM of A357 LPBF tensile samples: (**a**,**b**) as-built condition, (**c**,**d**) after stress-relieving, (**e**,**f**) after T6_B,170,3_.

**Table 1 materials-13-01488-t001:** Main process parameters adopted for A357 specimens.

Parameters	Values
Scanning speed v [mm/s]	600, 700, 800, 1000, 1200
Hatching distance h_d_ [mm]	0.10, 0.17, 0.20

**Table 2 materials-13-01488-t002:** Combination of process parameters, productivity, and relative density of A357 samples.

Sample	v [mm/s]	h_d_ [mm]	Prod_LPBF_ [mm^3^/s]	Relative Density [%]
1	800	0.20	4.80	99.39
2	1000	0.10	3.00	99.54
3	1200	0.20	7.20	96.00
4	1000	0.20	6.00	97.92
5	800	0.17	4.08	99.78
6	1000	0.17	5.10	99.17
7	800	0.10	2.40	99.81
8	1200	0.17	6.12	98.08
9	1200	0.10	3.60	99.94
10	600	0.10	1.80	99.45
11	600	0.17	3.06	99.09
12	600	0.20	3.60	99.36
13	700	0.10	2.10	99.16
14	700	0.17	3.57	99.17
15	700	0.20	4.20	98.97

**Table 3 materials-13-01488-t003:** A357 micro-hardness mean values for samples in different conditions.

LPBF A357 Specimens	HV_0.1_	S.D.
As-built	119	2
Stress-relieved	80	2
Stress-relieved + T6_B,170,3_	116	2

**Table 4 materials-13-01488-t004:** Tensile properties of A357 specimens as-built by LPBF and after two heat treatments.

Specimens	R_p, 0.2_ [MPa]		UTS [MPa]		ε [%]	
	Mean	S.D.	Mean	S.D.	Mean	S.D.
As-built	245	4	386	4	5.2	0.4
Stress-relieved	189	3	288	5	8.2	0.9
Stress rel + T6_B,170,3_	249	9	307	10	5.1	0.3

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
