# Peer review of "A357 Alloy by LPBF for Industry Applications"

_materials, 2020, doi:10.3390/ma13071488_

Round 1
Reviewer 1 Report
The aluminium alloys are very attractive for aerospace and automotive applications especially due to their lightweight. Nevertheless, their diminished hardness and low strength is still an issue. There are many published papers on this subject, but it seems that it still draws the attention of many researchers. From this point of view this paper is not a novelty, the authors themselves reporting in the past some results on the same subject on identical alloy A357 [14][49], and even on the same additive manufacturing technique based on laser powder bed fusion [14].
Evidently, the cited reference [14] used almost the same material and methods.
Nevertheless, the current research presents new interesting aspects on the A357 alloy. The paper reported experimental research and results on A357 alloy. Firstly, the process parameters are optimized from the viewpoint of productivity and density of obtained samples. Various samples were obtained for microstructural, hardness and tensile strength investigation in different conditions: as built, after a stress relieving treatment, and after a T6 precipitation hardening treatment carried out at different time and temperatures for solutioning and ageing.
I consider that this paper can be improved, so I have some suggestions for the authors.
1. At the end on Introduction section, the authors should emphasize the novelty of the current research in comparison with their own already published results and in relation to other previously published papers on connected subject. In the actual form, the degree of novelty presented by this research is not clear.
2. The results of XRD analysis have to be presented in detail, as the authors stated that they carried out also XRD analysis on the three LPBF A357 specimens: as-built; stress relieved, and stress relieved + T6B,170,3.
3. The exact results of XRD analysis are requested by some contradictory assertions found at lines 208-209: "On the other hand, XRD spectra did not reveal any peak for Mg2Si intermetallic compound." and those presented in Figure 6b (Mg2Si precipitates).
4. Some phrases are not very clear, so I advise the authors to recheck the entire paper. Some small errors ought to be amended, e.g., "AS expected", line 126: "[16,54 ïƒ era la 53] ", line 130: "[6,44,46, 55 ïƒ era la 54] " etc.
5. The conclusions should be rearranged without bullets, and in a clearer manner.
Author Response
The aluminium alloys are very attractive for aerospace and automotive applications especially due to their lightweight. Nevertheless, their diminished hardness and low strength is still an issue. There are many published papers on this subject, but it seems that it still draws the attention of many researchers. From this point of view this paper is not a novelty, the authors themselves reporting in the past some results on the same subject on identical alloy A357 [14][49], and even on the same additive manufacturing technique based on laser powder bed fusion [14].
Evidently, the cited reference [14] used almost the same material and methods.
Nevertheless, the current research presents new interesting aspects on the A357 alloy. The paper reported experimental research and results on A357 alloy. Firstly, the process parameters are optimized from the viewpoint of productivity and density of obtained samples. Various samples were obtained for microstructural, hardness and tensile strength investigation in different conditions: as built, after a stress relieving treatment, and after a T6 precipitation hardening treatment carried out at different time and temperatures for solutioning and ageing.
I consider that this paper can be improved, so I have some suggestions for the authors.
- At the end on Introduction section, the authors should emphasize the novelty of the current research in comparison with their own already published results and in relation to other previously published papers on connected subject. In the actual form, the degree of novelty presented by this research is not clear.
Thank you for your valuable advice. Now the Introduction Section was modified. Main differences with respect to other studies of other research groups in literature are related to the use of a different LPBF system. In fact it is known that each LPBF machine, even employing a similar laser source in terms of power, differ in a lot of other parameters, like scanning strategy, layer thickness, etc. Therefore even considering to start from the same powder, the samples obtained differ in terms of microstructure, as they experienced a different thermal history, and in terms of final mechanical properties. Considering the reference [14], a previous study on A357 alloy was performed. But in this case the parameters were chosen and fixed; the accent was put on changing the building platform temperature, and in evaluating the effects of some thermal treatments, like direct aging.
The focus of the present study was strictly connected to industrial needs as the title emplies. So it was decided to study systematically the effect of different process parameters, with the aim of reaching the highest density with the highest productivity. Furthermore the stress relieving treatment was considered a fixed point, fundamental for industrial needs. In fact in the production of a complex part, the real advantage given by AM technologies, fundamental is to release the thermal stresses before detaching it from the building platform. Only when detached it is possible to put the component in furnace and in the cooling media for the thermal treatment. In many cases, it would be difficult or impossible to treat the component still attached to the building platform. Moreover, detaching the part from the building platform allows to evacuate efficiently the residual powder entrapped during the process. But if you remove a complex part without a stress relieving, often you can have problems like distortion or simply loss of dimensional tolerances. Therefore also the systematic study of a possible T6 treatment to reach the highest hardness and strength was performed after the stress relieving treatment.
- The results of XRD analysis have to be presented in detail, as the authors stated that they carried out also XRD analysis on the three LPBF A357 specimens: as-built; stress relieved, and stress
Thank you for your suggestion. The answer is given at point 3.
- The exact results of XRD analysis are requested by some contradictory assertions found at lines 208-209: "On the other hand, XRD spectra did not reveal any peak for Mg2Si intermetallic compound." and those presented in Figure 6b (Mg2Si precipitates).
Dear reviewer, thank you for comments 2 and 3. Now new figure 6 shows the results of XRD analysis as different spectra.
- Some phrases are not very clear, so I advise the authors to recheck the entire paper. Some small errors ought to be amended, e.g., "AS expected", line 126: "[16,54 à era la 53] ", line 130: "[6,44,46, 55 à era la 54] " etc.
Thank you for your valuable advice. The paper was fully revised.
- The conclusions should be rearranged without bullets, and in a clearer manner.
Thanks for your kind suggestion. The conclusions were rearranged.
Reviewer 2 Report
The scientific level of paper is good, objectives are clearly stated and all conclusions are supported by the research methods presented and described in detail in the paper, but there are some editing (spelling and grammar) errors in the text that the authors are requested to correct them before publication, as indicated below:
- There is one term referring to one parameter that was varied in the process of SLM, which in few places is presented as “scan speed” or “scanning speed”. The right term is “scanning speed” and in order to be consequent in the text, “scan speed” term must be replaced by “scanning speed” wherever this expression occurs in the text (Line 56, Line 102, Line 157
- In line 48, the expression “While, parts with complex geometry were” must be replaced by “Parts with complex geometry are”…In this way only, the sentence makes sense. In line 91, the expression “While the graph of Figure 1b illustrates” must be replaced by “The graph of Figure 1b illustrates” from the same reason.
- Graphs from Figure 1b must contain a description on the ordinate display. If on the abscissa is given the Frequency [vol %], on the ordinate is shown what? The grain size in microns? Or??
- Expressions such as “a constant layer thickness”, “a laser power” (Line 103) must be given without “a” letter in front of the terms. These expressions must be given as “constant layer thickness”, “laser power”, etc. Same observation is valid for Line 140 “a Leica WHMT” and Line 142 “a X-Pert Phillips”…The expressions must be given without using letter “a” in the front. In Line 212 term “a homogenous” must be used without letter “a” in the front also. Same observations for Line 228 for the term “a mean”, Line 231 “ a fine and a coarse”, Line 232: “a heat”; Line 258: “a fine”, “a plastic”; Line 270: “a relative”; Line 271: “a production”; Line 272: “A stress”; Line 274: “a subsequent”, Line 280 “a more”, etc. In these entire cases, letter “a” must be deleted in the front of the used expressions.
- Line 105: “of67” must be replaced by “of 67”.
- I do not understand the term “era la” which occurs many times in the text when references are indicated (e.g. in line 115 – “[53 –era la 55]. Why references are not indicated as [53-55] or [53,55] in this case? Same observation is valid also for Lines 126 and 130.
- In Line 116, where “t” letter is indicated in the equation it is not very clear to which parameter “t” is referring to? This term is referring to the layer thickness? I think this explanation must be given in text, right after the equation is being presented.
- Expression “samples hardness” must be replaced by “the hardness of samples” (Line 138)
- Expression “of wanting to get parts” must be replaced by “of getting parts” (Line 161)
- In Lines 145, 147 and 149 term “are” must be replaced by “were”; in Line 162 term “is” must be replaced by “was”. Same observation is valid also for Line 274.
- Line 168: the expression “do not affect” must be replaced by “to not affect”.
- Line 173: the expression “are then summarized” must be replaced by “are summarized”. Term “AS” must be replaced by “As” in the same Line. Same observation is valid also for Line 240.
- Line 178: the expression “the 5h ones” must be replaced by “the ones aged for 5h”
- Line 184” the term “describes” must be replaced by “described”
- Lines 186 and 187: the expression “A357 fine Si particles and Mg precipitates should act as” must be replaced by “A357, fine Si particles and Mg precipitates, acting as”
- Line 189: term “reduces” must be replaced by “is reduced”
- Line 197: the expression “treatmentstraditionally” must be replaced by “treatments traditionally”
- Line 209: term “compund” must be replaced by “compound”
- Line 241: term “yield” must be replaced by “yield strength”
- Line 242: term “aoubt” must be replaced by “about”
- Line 266: The number of chapter is wrong. Number of Chapter must be 4, not 5 in this case.
- Lines 296, 359, 428, 431, 436: The year number must be given with “Bold” capital letters, in order to have the same style of presenting as the one of the other references given.
Author Response
Dear reviewer,
Thank you for your valuable comments. The Authors find them very helpful for revising and improving the quality of the paper. The authors followed all the suggestions.
Reviewer 3 Report
Do all changes, please

Author Response
Dear reviewer, thank you for your careful work. The machine used in this study is an EOSINT M270 Dual Mode, equipped with a 200W Yb fiber continuous laser beam with a nominal focus diameter of 100 μm (88 µm measured) as described in the section “Materials and method”. EOS tradename to indicate this process is Direct Metal Laser Sintering. It is the same process known also as Selective Laser Melting (SLM), or Laser Powder Bed Fusion (LPBF), as defined by ISO/ASTM 52900 standard. The authors agree that that laser cladding is an additive process highly anisotropic, but it is a complete different technology. The main differences among laser cladding and LPBF are: in laser cladding, the powder is co-axial with the energy source, the powder has a mean diameter of hundreds of µm, and there is just a gas shielding; in LPBF the powder is spread onto a building platform, it is very fine with a mean diameter of 30-40 µm, and the process is conducted in a chamber with inert atmosphere.
Considering Hipping, it is a very expensive process and it is commonly used for nickel based superalloys, which could be difficult to weld without cracks ). Hipping allows to close cracks and open porosities that still remain after the additive process. On the other hand, it can be stated that HIP treatmentis not used commonly for aluminum alloys. In the present study the Authors obtained full dense A357 alloy samples without cracks and therefore HIP treatments is not necessary.
If the state of the art has some gap about laser cladding and Hipping the reason is that these technologies are not related with the one employed in this manuscript (LPBF).
The Authors agree that the effect of machining on tensile samples could have an influence on the mechanical properties, even if they are static. But the aim of this study was to evaluate the effects of different thermal treatments with respect to the as built condition of machined samples after LPBF. Therefore all tensile specimens were machined in the same way.
The papers you cited are very interesting, but they are not related with the aim of this manuscript.
Round 2
Reviewer 1 Report
The paper was entirely revised considering all the suggestions.